

# Annual cycle of water vapour in the lower stratosphere over the Indian Peninsula derived from Cryogenic Frost-point Hygrometer observations

Maria Emmanuel[1, 3], Sukumarapillai V. Sunilkumar *[1], Muhsin Muhammed[1], Buduru Suneel Kumar[2], Nagendra Neerudu[2], Geetha Ramkumar[1], Kunjukrishnapillai Rajeev [1], Krishnasamyiyer Parameswaran[1].

[1]Space Physics Laboratory, Vikram Sarabhai Space Centre, Trivandrum-695022, India

[2]Tata Institute of Fundamental Research Balloon Facility, Hyderabad- 500062, India

[3]Department of Physics, University of Kerala, Trivandrum- 695034, India

*Correspondence to: Sunilkumar S V (sv_sunilkumar@vssc.gov.in)

**Abstract:** In situ measurements of lower stratospheric water vapour employing Cryogenic Frost point Hygrometer (CFH) over two tropical stations, Trivandrum (8.53 °N, 76.87 °E) and Hyderabad (17.47 °N, 78.58 °E) over the Indian subcontinent are conducted as part of Tropical Tropopause Dynamics (TTD) monthly campaigns under GARNETS program. The annual variation of lower stratosphere (LS) water vapour clearly depicts the so called tape recorder effect at both the stations. The ascent rate of water vapour compares well with the velocity of Brewer-Dobson circulation and is slightly higher over the equatorial station when compared to the off-equatorial station. The column integrated water vapour in the LS varies in the range 1.5 to 4 g/m$^2$ with low values during winter and high values during summer monsoon and post monsoon seasons and its variability shows the signatures of local dynamics. The variation in water vapour mixing ratio (WVMR) at the cold point tropopause (CPT) exactly follows the variation in CPT temperature. The difference in WVMR between the stations shows a semi-annual variability in the altitude region 18-20 km region with high values of WVMR during summer monsoon and winter over Hyderabad and during pre-monsoon and post-monsoon over Trivandrum.  This difference is related to the influence of the variations in local CPT temperature and deep convection. The monsoon dynamics has a significant role in stratospheric water vapour distribution in summer monsoon season.

## 1 Introduction

Water vapour, a prominent greenhouse gas in the Earth's atmosphere plays a pivotal role in climate change (e.g., Forster and Shine, 2002). The amount of water vapour in the stratosphere is of great importance because it is the main source of hydroxyl radicals influencing the ozone abundance in this region and it contributes significantly to the radiative balance of



the region (Kley et al., 2000; Fueglistaler et al., 2009;). Variability in the distribution of stratospheric water vapour has important implications on both stratospheric and surface temperatures (Solomon et al., 2010; Wright et al., 2011). Due to the large residence time (of more than a year) stratospheric water vapour contributes significantly to the climate forcing instead of a simple response (Wang et al., 2009). Its feedback in a chemistry climate model is estimated to be +0.3 $Wm^{-2}K^{-1}$; one

third of which is contributed by the increase in water vapour entering the stratosphere through the tropical tropopause layer and the rest from the increase in water vapour entering through the extra tropical tropopause (Dessler et al., 2013).

    The prime processes responsible for water vapour in the stratosphere are slow ascent from troposphere, large scale motion and turbulent diffusion through tropopause, rapid ascent by convection or volcanic eruption and oxidation of methane in the upper stratosphere (Danielsen, 1993; Brasseur et al., 1999; Dessler and Sherwood, 2004; Rohs et al., 2006;

Wang et al., 2009). The amount of water vapour in the tropical upper troposphere and lower stratosphere (UTLS) region is highly dependent on the freeze drying in the tropical tropopause layer (Fueglistaler et al., 2009) and transport in the diabatic meridional Brewer Dobson circulation (BDC) (eg., Holton et al., 1995; Fueglistaler et al., 2005). Its seasonal variation in the tropical lower stratosphere (LS) region is linked to the seasonal variation in tropical tropopause temperature and is often called the tape recorder signal which was first discovered from the observations of Halogen Occultation Experiment

(HALOE) and Microwave Limb Sounder (MLS) on board Upper Atmosphere Research Satellite (UARS) (Mote et al., 1996; Randel et al., 2001). Rosenlof (2002) depicted the role of microphysical processes in regulating the amount of lower stratospheric water vapour and thereby negating the direct relationship between the tropical tropopause temperature and lower stratospheric water vapour. El-Nino Southern Oscillation (ENSO), planetary waves and Quasi-Biennial Oscillation (QBO) can cause variations in the tropical lower stratospheric water vapour by regulating the tropical tropopause

temperature and ascending speed of BDC (Fujiwara et al., 2001; Niwano et al., 2003; Randel et al., 2004; Fueglistaler and Hayes, 2005; Suzuki et al., 2013).

    Several studies suggest convection over the South East Asian monsoon region as the prime contributor for the wet phase of the stratospheric water vapour tape recorder effect (Bannister et al., 2004; Lelieveld et al., 2007). While the stratospheric water vapour in the monsoon regions is thought to be controlled by the large scale circulation and temperature

(Randel et al., 2015) the sub-seasonal variability is significantly higher over Asian monsoon region due to the peculiar dynamical features prevailing over the Indian sub-continent. This includes the largest annual migration of inter tropical convergence zone (ITCZ), SST exceeding 30 ℃ in the 'Arabian Sea Warm Pool' region during the April-May period, stronger and deeper convective clouds over the Bay-of-Bengal, monsoon circulation and Asian anticyclone prevailing over the upper troposphere (Randel et al., 2015) during summer-monsoon period and the high occurrence of thick cirrus clouds

(James et al., 2008; Devasthale and Fueglistaler, 2010; Meenu et al., 2011).



Though satellite borne measurements (eg., MLS, HALOE) have provided an overall picture of the water vapour variability in the stratosphere, accurate in situ measurements of water vapour in the UTLS region is rather scarce because of the technical difficulties in measuring such low amount of water vapour at low temperatures prevailing in those altitudes. Though accurate measurement of water vapour using conventional radiosondes is confined upto an altitude of 10 km,

balloon borne frost point hygrometers are capable to measure water vapour from the surface up to an altitude of ~28 km (Mastenbrook, 1962, 1968, Vomel et al., 2007). Measurement of stratospheric water vapour using frost point hygrometers began in 1960's and the longest dataset is available over Boulder, Colorado (40 °N, 105 °W) (Oltmans et al., 2000; Scherer et al., 2008; Hurst et al., 2011). Fujiwara et al. (2010) studied the seasonal and decadal variability of lower stratospheric water vapour over the tropics using balloon borne cryogenic frost point hygrometer data obtained during different campaigns

in the period 1993-2009 over the Pacific, South-East Asia and Costa Rica. Several campaigns conducted in the tropical regions have also addressed the variability of the UTLS water vapour using frost point hygrometers.

Over India, launches of improved version of the cryogenic frost-point hygrometer (CFH) are being carried out since April 2014 from Trivandrum (8.53 °N, 76.87 °E) and Hyderabad (17.47 °N, 78.58 °E) as a part of Tropical Tropopause Dynamics (TTD) experiments under GARNETS (GPS Aided Radiosonde Network Experiment for Troposphere stratosphere

Studies) program (Sunilkumar et al., 2016). Trivandrum is an equatorial coastal station on the south west coast of Indian peninsula and Hyderabad is an off equatorial inland station. In this paper, we discuss the in situ observations of annual and seasonal variations of lower stratospheric water vapour over these two stations during the period 2015-2016. A detailed study of water vapour in the lower stratosphere using in situ data from CFH is first of its kind over Indian Peninsula.

## 2 Experiment and data

The balloon-borne Cryogenic Frost point hygrometer (CFH) is a micro-processor controlled light weight instrument which accurately measures the frost point temperature in the 0-28 km altitude region. Description of the instrument which work based on chilled mirror principle is detailed in Vomel et al. (2007a). The frost-point temperature measured by CFH is used to estimate the vapour pressure using Goff-Gratch equation (Goff and Gratch, 1946) and then estimate the water vapour mixing ratio (WVMR) (Vomel et al., 2002, 2007a) and absolute humidity. The total uncertainty of frost-point temperature is

approximately 0.51 °C. This leads to an uncertainty in the estimated WVMR of about 4 % in the lower troposphere, 9 % in the tropopause region and 10 % in the lower and middle stratosphere (Vomel et al., 2007a). During the campaigns, the CFH is interfaced with an Electrochemical Concentration Cell (ECC) Ozonesonde and iMet-1 radiosonde. The payload is attached to a meteorological balloon weighing ~2 kg which attain an average altitude of ~30 km before the balloon burst. The accuracies of iMet-1 radiosonde measured temperature and pressure are 0.3 °C and 0.5 hPa respectively (Hurst et al., 2011).

Twenty four balloon soundings from Hyderabad (two in a month) and twelve from Trivandrum (one in a month) carried out





during the one year period February 2015 to January 2016 are used in this study. Raw data having a vertical resolution of 4-6 m are re-gridded to a uniform vertical resolution of 100 m mainly to avoid the random fluctuations due to balloon oscillations.

The MLS uses 190 GHz band to measure water vapour in the upper troposphere and stratosphere region. It provides ~3500 profiles daily at a latitude interval of ~1.5° from surface to an altitude of 95 km. The data useful for scientific purposes is available from 316 hPa (~8 km) to 0.1 hPa (~64 km) (Schoeberl et al., 2006; Lambert et al., 2007; Read et al., 2007). MLS provides water vapour profiles with a vertical resolution of 2-3 km, 4-6 km and 8 km at 316 hPa to tropopause, tropopause to 1 hPa and at 0.1 hPa with precisions of ~15 %, ~0.1 ppmv and ~0.3 ppmv respectively (Waters et al., 2006). The accuracy of MLS retrieved water vapour mixing ratio values varies from 4 % to 8.3 % in the lower stratosphere region

(100-18 hPa) (Livesey et al., 2013; Hurst et al., 2014). The version 4.2 of WVMR profiles in a latitude-longitude grid of 5° x 10° centred around the launch sites during the period 2009-2015 are used in the present study along with the CFH measurements to study the features of stratospheric water vapour.

COSMIC-RO wet profiles provide temperature with a vertical resolution of 100 m. The RO profiles are found to be quite accurate within ±1 K in the altitude range 5-20 km (Ware et al 1996; Kurisinski et al 1997; Staten and Reichler, 2008).

Anthes et al. (2008) have estimated the temperature precision of COSMIC-RO to be ~0.25 K between 10 and 20 km. In the present study, COSMIC-RO derived temperature profiles in the latitude longitude grid of 5° x 5° around the stations for the period 2011- 2015 are used to determine the temperature and altitude of cold point tropopause (CPT). Very High Resolution Radiometer (VHRR) on board KALPANA-1 measures radiances in the atmospheric window of Thermal Infrared (TIR: 10.5-12.5 μm). The brightness temperatures (BT's) derived from these radiances are used to infer the cloud top altitudes, which is

used as a proxy for convection. If the BT's in the TIR band is less than 220 K, it can be inferred that deep convective cloud tops reaches an altitude greater than 12 km (Roca et al., 2002; Rajeev et al., 2008). In the present study, TIR BT's in a spatial grid of 0.6° x 0.6° around the balloon launch sites during the period 2011-2015 are used to infer the occurrence frequency of very deep convection (expressed as percentage). The vertical wind in a latitude- longitude grid of 0.5° x 0.5° around the launch site during the period 2011-2015 obtained from the ERA (ECMWF reanalysis)-interim is used in the present study to

delineate the annual variation in the vertical wind over both the stations. Even though, the amplitude of vertical wind derived from reanalysis may be not that accurate, the wind direction will be reliable to infer updrafts and downdrafts. As the vertical wind is expressed in units of Pa/s, negative value represents an upward motion and positive value a downward motion.



### 3 Results and discussions

#### 3.1 Altitude distribution of lower stratospheric water vapour and its seasonal variation

Generally, water vapour mixing ratio decreases with altitude in the troposphere reaching a minimum value near the tropopause and remains almost steady in the lower stratosphere region. Figure 1 show typical profiles of temperature and
WVMR over Hyderabad and Trivandrum during the two contrasting seasons, summer (14 July 2015 over Trivandrum and 27 July 2015 over Hyderabad) and winter (31 December 2014 over Trivandrum and 23 December 2014 over Hyderabad). The ambient temperature is higher in the summer profile than in winter profile in the altitude region 12 to 20 km while it is almost the same in profiles in both the seasons above 20 km over Hyderabad. At Trivandrum, the altitude region 12 to14 km and above 17.5 km is warmer and the altitude region 14 to 17.5 km is cooler in summer profile when compared to the winter
profile. At Trivandrum, the CPT altitude (temperature) is ~ 17.3 km (~190 K) in the summer profile and 18.3 km (~192 K) in the winter profile while at Hyderabad, the CPT altitude (temperature) is 16.9 km (198 K) in the summer profile and is 18.5 km (~ 193 K) in the winter profile. This shows that the CPT is cooler for the summer profile (14 July 2015) at Trivandrum, which is inconsistent with the climatological mean value for that season. The cooler CPT on this day could be due to the influence of deep convection and the associated wave generation. KALPANA-1 VHRR observations (figure not shown)
show the presence of deep convection on and before the day of observation. Wind and temperature field analysis (figure not shown) shows the persistence of waves in the UTLS region on this day. There were several studies showing the influence of deep convection and waves on the temperature and water vapour in the UTLS region (eg: Suzuki et al., 2013; Sunilkumar et al., 2016; Muhsin et al., 2018). This cooler tropopause might be responsible for the freeze drying around this region resulting in a sharp minimum in WVMR close to the CPT.

In the upper troposphere from 14-16 km, the WVMR shows relatively higher values in summer profile over both the stations, associated with the deep convection in summer monsoon season. While the WVMR decrease almost steadily in this altitude region in both the summer and winter profiles over Hyderabad, it shows a small peak of ~20 ppmv around ~15 km in the summer profile over Trivandrum. The minimum in the WVMR, called the hygropause occur just above or below the CPT. While the hygropause at both the stations occurs above the CPT in summer profiles, it occurs below the CPT in winter
profiles. The WVMR at hygropause is around 2 ppmv over Trivandrum and is around 3- 3.5 ppmv over Hyderabad. The hygropause is found to be sharper with lesser WVMR over Trivandrum compared to Hyderabad. The mixing ratio at the hygropause is less in the winter profile over Hyderabad while it is less in the summer profile over Trivandrum. Low CPT temperature at Trivandrum (< 191 K) in the summer profile is conducive for freeze drying and could have led to a drier and sharper hygropause during summer. In the lower stratosphere, while the WVMR is higher in winter than summer in the




altitude region 18- 21 km, it shows an opposite behaviour in the altitude region above 21 km. This feature is common for both the stations.

Figure 2 shows the seasonal mean profiles of lower stratospheric water vapour mixing ratio in all seasons over both the stations. The WVMR value varies from 2- 6 ppmv in the LS region (16- 25 km) over Trivandrum and Hyderabad. Generally, it increases with altitude from a minimum value of 3- 5 ppmv around 16- 17 km in the lower stratosphere region. But the altitudinal structure of the lower stratospheric water vapour profile is different in different seasons. During winter (December, January and February; DJF) WVMR shows a prominent minimum at ~17- 18 km with a value of ~3.5 ppmv (~2.8 ppmv) over Hyderabad (Trivandrum) and a secondary minimum at ~22 km with WVMR around ~4 ppmv over both the stations. The maximum in WVMR is observed around ~19.5 km with a value of ~4.5 ppmv. The minimum in WVMR at ~18 km (~2.7 ppmv) during pre-monsoon (March, April and May; MAM) propagates upward to an altitude of ~19.5 km (~3.5 ppmv) during the summer monsoon season (June, July, August and September; JJAS) at both the stations. Over Trivandrum, the seasonal mean WVMR profile shows a broad minimum in the altitude region 19-21 km during summer monsoon; this is mainly due to the large month-to-month variability of WVMR which might be the result of synoptic variabilities in this altitude region from June to September period. During the post monsoon (October and November; ON) period WVMR shows a maximum at ~18.5 km (~5 ppmv) and a minimum at ~20.5 km (3.5- 4 ppmv). The water vapour that enters the lower stratosphere propagates upward by means of BDC which has an average ascent rate of 0.2 mm/s. Thus, the ascent rate of lower stratospheric water vapour could be taken as a proxy for the ascent rate of the BDC over the two stations. The average ascent rate of water vapour in the LS estimated from the lead lag correlation of the mean WVMR profiles in DJF and JJAS from 16 to 22 km shown in Figure 2 is 0.11 mm/s over Hyderabad and 0.14 mm/s over Trivandrum.

## 3.2 Annual variation in lower stratospheric water vapour and its causative mechanisms

Annual variation in the tropical lower stratospheric water vapour follows the annual cycle in the effective entry value of the water vapour to the LS regulated by the annual variation in the tropical tropopause temperature (Holton et al. 1995; Mote et al., 1996; Fujiwara et al., 2010). The cold point tropopause is situated at a higher altitude and is colder in winter with respect to the summer season. Hence, the amount of water vapour entering the LS region is relatively less (~2- 3 ppmv) in winter and more in summer (~4- 6 ppmv). The annual cycle of the lower stratospheric water vapour also shows a phase shift with altitude due to the slow upward propagation of water vapour with the BDC and is usually called the tape recorder effect. Figure 3 shows the month to month variation of altitudinal structure of the lower stratospheric water obtained using CFH at both stations during the period February 2015- January 2016 depicting the annual cycle. Even though the number of profiles were less, the annual cycle of WVMR profiles show a clear signature of tape recorder effect over both the stations, with high



value of WVMR propagating from an altitude around 16.5 (16) km in June to an altitude of 19.5 (19.5) km in December at Hyderabad (Trivandrum) and low value in WVMR (~2 ppmv) propagating from an altitude of 18.05 (18.05) km in February to at an altitude of 21.25 (21.95) km in December over Hyderabad (Trivandrum). The average ascent rate calculated from the upward propagation of the minimum in WVMR from February to December over Hyderabad is ~0.12 mm/s (10.56 m/day) and that over Trivandrum is ~0.14 mm/s (12.9 m/day). The estimated ascent rate over Hyderabad is slightly lower than that over Trivandrum.

The column integrated water vapour in the lower stratosphere ($IWV_{LS}$) estimated by integrating the absolute humidity in the altitude region from CPT to 25 km represents the actual amount of water vapour in this region. As the absolute humidity decreases almost exponentially with altitude, the variability in $IWV_{LS}$ will be mainly representing the variability of water vapour in the altitude region just above the CPT. Figure 4 shows the monthly mean $IWV_{LS}$ for Trivandrum and Hyderabad. The $IWV_{LS}$ generally varies from 1.5- 4 g/m$^2$ at both the stations, which is about one in ten thousandth of the total atmosphere column-integrated water vapour in the troposphere. It is relatively high during summer than winter. Over Hyderabad, the $IWV_{LS}$ is the lowest in January (1.51 g/m$^2$) and attains the peak value in October (3.91 g/m$^2$). Such a clear annual pattern is not observed over Trivandrum. $IWV_{LS}$ is generally in the range 2.1 to 3 g/m$^2$ at Trivandrum except for the high values in June and October months. Though on an average the water vapour content is high at Hyderabad, instantaneous high values in $IWV_{LS}$ is observed at Trivandrum, at the beginning and ending of the summer monsoon season. Over Trivandrum, occurrence of deep convective clouds is high, mostly in the second half of May. The instantaneous high values in June over Trivandrum could be attributed to the input of high amount of water vapour by means of deep convection prior to the onset of the summer monsoon. High values observed in October over Trivandrum and Hyderabad is linked to the local deep convections. As the variability in $IWV_{LS}$ clearly shows the influence of local processes, the variability in local cold tropopause temperature and deep convection and their possible link to the variations in water vapour was examined in detail. Figure 5a shows the annual variation of CPT altitude and hygropause altitude over both the stations obtained from CFH-radiosonde observations. The CPT altitude varies in the range 16 to 18.5 km at both stations and is generally higher in winter and lower in summer. While the hygropause is situated at a lower altitude during January- May, it is at a higher altitude during June- November period. This pattern is almost opposite to that of CPT altitude. The difference between the hygropause and CPT altitude is more in summer and less in winter. Figure 5b shows the CPT temperature along with WVMR at the CPT and the hygropause. The CPT temperature varies between 187 K and 200 K over Trivandrum and it varies between 190 K and 194 K over Hyderabad. The WVMR at CPT varies between 2 and 6 ppmv and its monthly pattern strictly follows the CPT temperature. While the annual pattern of WVMR at hygropause is almost similar to that at CPT over Trivandrum, the hygropause WVMR is almost steady during July- February over Hyderabad.





As the water vapour transport in the LS region is a very slow process, the mean variability in CPT altitude and temperature is more important than the instantaneous values in determining the mean lower stratospheric water vapour distribution. Hence, the mean annual variation of CPT altitude (shown as horizontal lines in Figure 3) and CPT temperature (Figure 6a ) over both the stations are examined using the COSMIC-RO data during the period 2011- 2015. The monthly mean CPT altitude varies from 16.5 to 17.5 km over Trivandrum and from 16.9 to 17.8 km over Hyderabad with relatively higher altitude during winter and pre-monsoon season and lower altitude during summer monsoon and post-monsoon season. The CPT temperature shows an annual variation with a prominent peak in the summer monsoon and a secondary peak in the month of March. The amplitude of this annual cycle is high over Trivandrum compared to Hyderabad. The monthly mean CPT temperature varies from ~188 to 194 K over Trivandrum and varies from 190 to 193 K over Hyderabad. An important feature is the occurrence of very low value of CPT temperatures ($\leq$ 191 K) over both the stations which are conducive for freeze drying. For the prevailing water vapour concentrations near the tropopause (~3 ppmv) a value <191 K is favourable for triggering freeze drying mechanism (Newell and Gould-Stewart, 1981; Tsuda et al. 1994; Jain et al. 2011). Statistics shows that the occurrence frequency of CPT temperature <191 K is relatively higher (60-70 %) in winter and pre-monsoon compared to summer monsoon season (~20- 40 %) over the Indian peninsula. This indicates the higher probability for the occurrence of freeze drying in winter and pre-monsoon seasons and leads to a minimum value in WVMR (2-3 ppmv) around the tropopause region during these seasons.  In general the mean annual pattern in CPT temperature is consistent with the annual pattern of WVMR obtained from CFH observations at CPT (Figures 3 and 5) and this could explain the entry value of water vapour into the LS region and the tape recorder effect.

Another important factor which could determine the amount of lower stratospheric water vapour is the occurrence of local deep convection reaching the tropical tropopause layer (with deep convective cloud tops >12 km) which could directly pump water vapour to the tropical tropopause and lower stratosphere and destabilize the CPT directly or through triggering of atmospheric waves (eg., Jensen and Pfister, 2004). The occurrence frequency of deep convection is examined using TIR brightness temperature data obtained from KALPANA-1 VHRR over both the stations. The monthly mean occurrence frequency of deep convections with deep connective cloud tops >12 km (TIR BT < 220 K) over both the stations Trivandrum and Hyderabad is shown in Figure 6b. The occurrence frequency of deep convection is relatively high during summer monsoon season at Hyderabad and during pre-monsoon and post-monsoon season at Trivandrum. During summer monsoon, Tropical Easterly Jet (TEJ) is very strong in the upper troposphere and its core is situated very close to CPT. In addition, the monsoon anticyclone dynamics and long range transport of water vapour from the deep convective sources over the Bay of Bengal (BoB) also play a role in determining the distribution of water vapour in the LS during the summer monsoon season. During this season, strong anticyclonic circulation and divergence persist in upper troposphere north of 15 °N and is linked to the deep convection over South East Asia. This circulation plays a major role in the confinement of high





amount of water vapour in the upper troposphere and its quasi isentropic vertical transport by means of large scale motion (eg: Randel and Park, 2006; Park et al., 2007). Hyderabad being closer to the deep convective core situated over the BoB and being situated at the fringe of the monsoon anticyclonic circulation, the advection of moisture during summer monsoon season in to the UTLS region over this station is marginally higher compared to Trivandrum (Figures 2 and 4).

The monthly variation of vertical wind derived by ERA-interim in the altitudinal region 10- 20 km over Hyderabad and Trivandrum for the period 2011-15 is shown in Figure 7. Though the magnitude of vertical wind in any reanalysis may not be very accurate, the direction (updraft/ downdraft) would be quite reliable. In the upper troposphere, the vertical wind shows a strong updraft during the summer monsoon season (JJAS) and a weak downdraft in all other seasons over Hyderabad.  The vertical wind over Trivandrum shows updraft in the upper troposphere throughout the period from pre-
monsoon to post-monsoon seasons with relatively stronger updrafts during summer monsoon. The strong updraft extending up to the tropopause region might be responsible for the high amount of water vapour observed near the CPT (Figures 1- 4) during monsoon season at both stations. During the pre-monsoon (MAM) and post monsoon (ON) seasons the vertical wind is upward (updraft) associated with deep convection over Trivandrum. This leads to relatively higher value of water vapour near the CPT region in these seasons over Trivandrum.

**3.3 Spatial gradient in lower stratospheric water vapour over Trivandrum and Hyderabad sectors**

The amount of water vapour in the lower stratosphere over Trivandrum and Hyderabad do show considerable differences throughout the altitudes. Figure 8a shows the annual variation in the difference in WVMR between the two stations (Hyderabad - Trivandrum) observed from CFH measurements along with the annual variation of CPT altitude at both stations from COSMIC-RO data (2011-2015). Positive values indicate a higher mixing ratio over Hyderabad. The
difference in WVMR shows a seasonal variability with magnitude ranging from ~ -0.8 ppmv to +1.5 ppmv. This difference is beyond the uncertainty limit of the CFH in the LS region. As the mean CPT altitude is higher at Hyderabad with respect to Trivandrum in all the seasons the amount of water vapour is generally high at Hyderabad with respect to Trivandrum in the altitude region 16.5 to 17 km. The WVMR just above the CPT altitude, around 17-18 km, is relatively high over Hyderabad during summer monsoon (JJAS) and winter seasons (DJF) and high over Trivandrum during pre-monsoon (MAM) and post-
monsoon (ON) seasons.  This difference shows an upward propagation with time, similar to the tape recorder effect. The signature of this upward propagation is seen up to an altitude around 22-23 km. In order to confirm whether this difference is only due to changes in water vapour entering the lower stratosphere (difference in WVMR could be due to the difference in pressure also since it is the ratio of vapour pressure to the atmospheric pressure) the difference in absolute humidity over the two stations are also calculated and its annual variation is shown in Figure 8b. The absolute humidity difference also shows a
similar annual variability with amplitude varying from ~ -0.08 mg/m$^3$ to 0.15 mg/m$^3$.  This confirms that the observed





difference is due to variation in stratospheric water vapour density. In order to confirm the consistency of the observed semi-annual variability in the difference in water vapour density between the two stations the same difference for more number of years are examined using MLS data. The mean of WVMR within a spatial grid of 5° x 10° around the two stations Trivandrum and Hyderabad are used for this analysis assuming that the variability in lower stratospheric water vapour within

this grid is negligible. The inter-annual variability in WVMR difference between Trivandrum and Hyderabad grids is examined by subtracting the respective monthly mean value of WVMR in Trivandrum grid from that in Hyderabad grid. Figure 9 shows the inter-annual variability in WVMR difference between Trivandrum and Hyderabad grids for the period 2009 to 2015. The difference in WVMR derived from MLS observations also shows an upward propagating semi-annual pattern with WVMR higher over Hyderabad in summer monsoon and winter and higher over Trivandrum in the other two

seasons around 16-18 km altitude. The amplitude of difference in MLS WVMR ranges between ~ -0.2 and 0.6 ppmv with higher amplitude during the summer monsoon season. The amplitudes observed are smaller than that obtained by the CFH observations. Emmanuel et al. (2018) has reported a mixing ratio dependent bias for MLS over Trivandrum and Hyderabad and this could be the reason for the smaller amplitude of difference in WVMR obtained from MLS.

       From Figure 6a it is clear that while the CPT is relatively warmer in the summer monsoon season over Trivandrum, it

is warmer in winter over Hyderabad. If CPT temperature alone was the controller of the lower stratospheric entry of water vapour, the amount of water vapour must have been higher over Trivandrum during summer monsoon and higher over Hyderabad in all other seasons. But the assumption is only true in winter season when the mean CPT temperature is less than 191 K. As the CPT temperature is lower over Trivandrum, the amount of water vapour entering the stratosphere is relatively lower over this station with respect to Hyderabad in winter. But in summer monsoon season also the water vapour mixing

ratio is higher over Hyderabad indicating that the lower stratospheric water vapour difference is not related to the difference in CPT temperatures between the stations alone in this season. The other factor to be examined is the difference in occurrence frequency of deep convection. The relative occurrence of deep convection is higher over Hyderabad during the summer monsoon period and it is higher over Trivandrum during pre- monsoon and post monsoon season (Figure 6b). This could be responsible for the observed high value of WVMR above the CPT over Hyderabad during summer monsoon period

and over Trivandrum during the pre-monsoon and post monsoon periods. In summer monsoon season, the advection of water vapour from the deep convective BoB region and confinement within the anticyclone is also contributing the high in WVMR over Hyderabad. Thus even though the mean annual cycle of lower stratospheric water vapour is controlled by the CPT temperature with more water vapour during boreal summer and less water vapour during boreal winter the spatial variability in lower stratospheric water vapour is controlled by CPT temperature and freeze drying in the winter season and local deep

convective vertical transport during the other seasons. The long range advection also plays a significant role in the variability of lower stratospheric water vapour in summer monsoon season.



**4 Summary and conclusions**

The seasonal and annual variations of lower stratospheric water vapour over two tropical stations Hyderabad (South central India) and Trivandrum (South west Peninsular India) in the Indian peninsula have been investigated using water vapour mixing ratio derived from CFH observations of frost point temperature.

Generally, the WVMR decreases with altitude in the troposphere region and increases with altitude in the LS region with seasonal difference in the altitudinal structure. The seasonal variation of WVMR in the LS region over both the stations shows signatures of atmospheric water vapour tape recorder effect with two WVMR minima at ~17.5 km and 22 km in winter (DJF) and a single minimum at ~19.5 km during summer monsoon (JJAS). The ascent rate of water vapour in the lower stratosphere region over Hyderabad is slightly lower than that over Trivandrum. The column integrated water vapour

($IWV_{LS}$) in the lower stratosphere (CPT to 25 km) is generally 1.5- 4 $g/m^2$ over both the stations, which is about one by ten thousandth of the total atmosphere column-integrated water vapour. $IWV_{LS}$ is more during summer months than the winter months. Over Hyderabad the $IWV_{LS}$ shows a pronounced annual cycle with a minimum in January (1.51 $g/m^2$) and a maximum in the month of October (3.91 $g/m^2$). Over Trivandrum, the maximum occur in October and minimum in January and show a prominent peak in the month of June. The variations in $IWV_{LS}$ indicate the influence of local dynamics on the

water vapour in the lower stratosphere especially in the altitude region very near to the tropopause. The annual pattern of WVMR at the CPT directly follows the annual pattern in local CPT temperatures.

The difference in lower stratospheric CFH observed WVMR between Hyderabad and Trivandrum shows a semi-annual variation with more water vapour entering the lower stratosphere region over Hyderabad in summer monsoon and winter seasons and over Trivandrum in pre-monsoon and post-monsoon seasons. The similar difference pattern seen in the

absolute humidity values confirm that it is caused by the difference in real water vapour content entering the lower stratosphere region over these stations/regions. These features are seen in MLS derived WVMR also. The difference in CPT temperature over the stations could only explain the winter high over Hyderabad. The summer high in water vapour over Hyderabad and pre-monsoon and post-monsoon high in water vapour over Trivandrum seems to have originated from the deep convective sources pumping moisture from upper troposphere to the lower stratosphere region. The long range

advection of water vapour from the deep convective source and its confinement in the monsoon anticyclone is also playing a role in the summer monsoon season. Thus, the present study indicates that, though the mean annual cycle in lower stratospheric water vapour is determined by the annual cycle in the CPT temperature and large scale dynamics, the spatial difference in the annual cycle is influenced by the effect of local processes. The combined effect of CPT temperature and occurrence of deep convection and monsoon dynamics is clearly seen at and just above the cold point tropopause. The real

quantification of contributions of different factors to the variability of lower stratospheric water vapour variation over Indian



region needs to be carried out by further investigations with special emphasis on the relative role of local deep convection and long range transport.

**Acknowledgements**

CFH observations used for this study were conducted as a part of Tropical Tropopause Dynamics (TTD) experiments under the GARNETS (GPS Aided Radio- sonde Network Experiment for Troposphere-stratosphere Studies) of Space Physics Laboratory (SPL), Vikram Sarabhai Space Centre (VSSC). The authors are thankful to the technical and scientific staff of SPL and METF (Meteorological Facility) of VSSC and TIFR-BF (Tata Institute of Fundamental Research-Balloon Facility) for their dedicated efforts in conducting the CFH experiments. The Aura-MLS data was obtained from NASA through their website (https://disc.gsfc.nasa.gov/). COSMIC-RO temperature data was obtained from UCAR website (http://cdaac-www.cosmic.ucar.edu/cdaac/) and KALPANA-1 VHRR Brightness temperature data was obtained from MOSDAC website (www.mosdac.gov.in). ERA-interim vertical wind data was obtained from the ECMWF website (http://apps.ecmwf.int/datasets/). Maria Emmanuel and M. Muhsin were supported by the ISRO Research Fellowship.

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





**Figures**

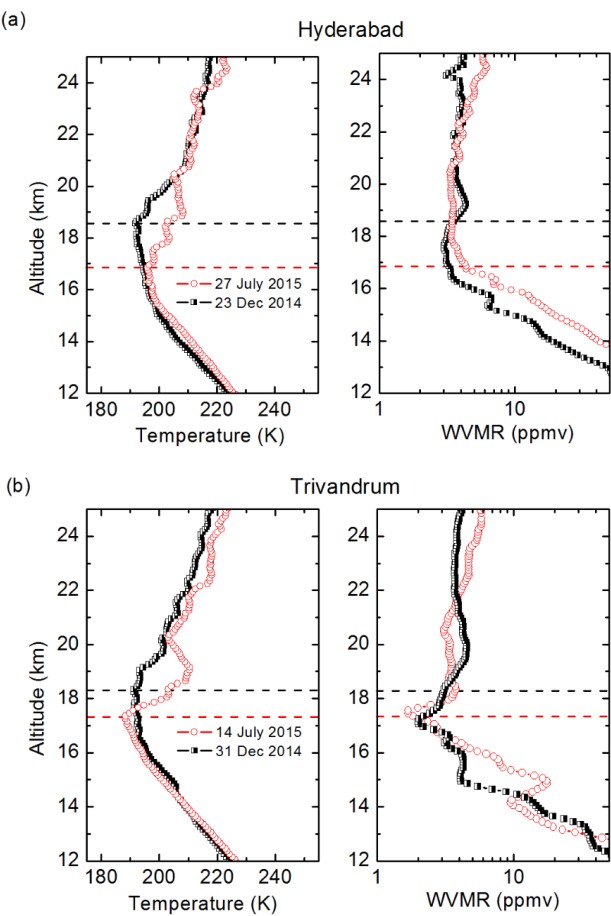

**Figure 1:** Profiles of ambient temperature and CFH observed WVMR in two contrasting seasons, summer monsoon and winter over (a) Hyderabad (b) Trivandrum. The horizontal lines represent the corresponding cold point tropopause (CPT) altitudes.



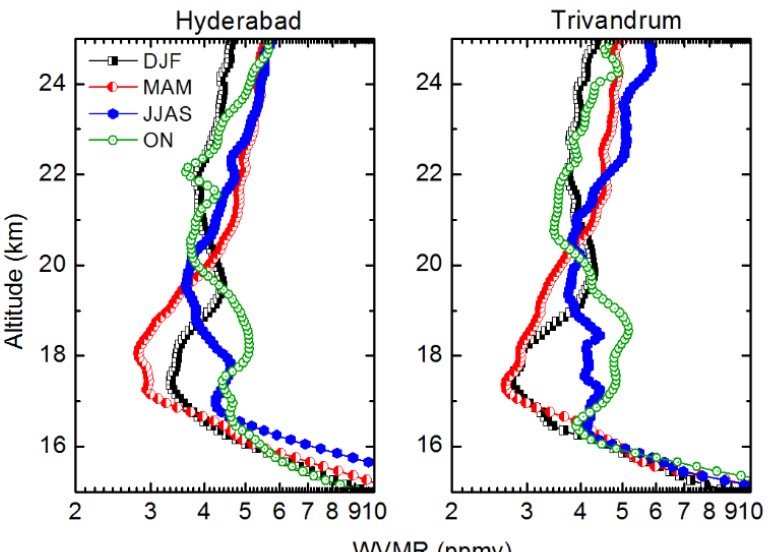

**Figure 2:** Seasonal mean profiles of CFH observed WVMR in the lower stratosphere over Hyderabad and Trivandrum in winter (DJF), pre-monsoon (MAM), summer monsoon (JJAS) and post-monsoon (ON) during the period February 2015- January 2016.




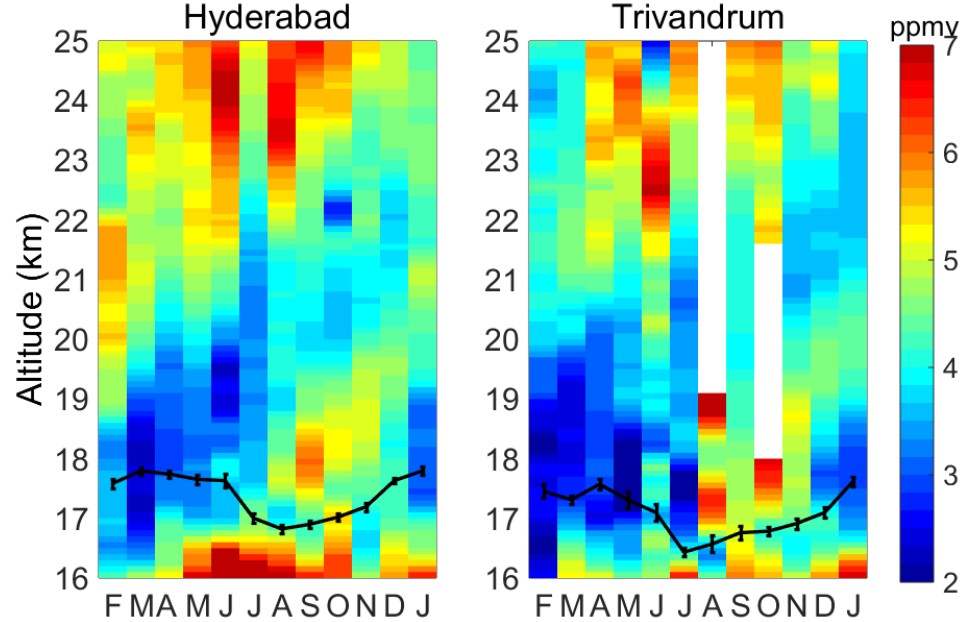

**Figure 3:** Annual cycle of CFH obtained lower stratospheric water vapour mixing ratio over Hyderabad and Trivandrum during the period February 2015- January 2016. The horizontal line represents the mean CPT altitude from COSMIC-RO data during the period 2011-2015.





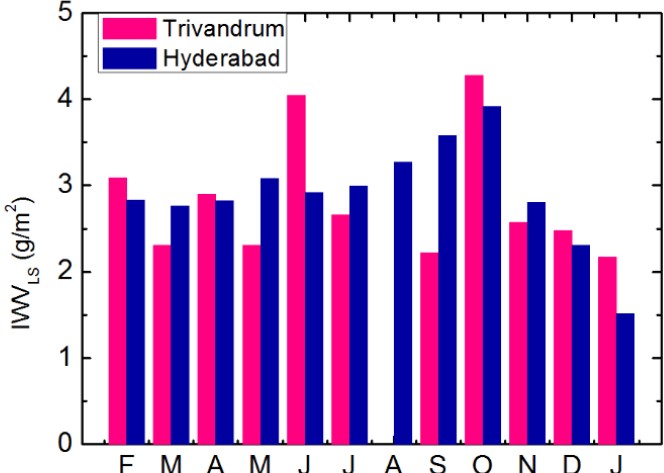

**Figure 4:** Annual cycle of column integrated water vapour in the lower stratosphere ($IWV_{LS}$) during the period February 2015 - January 2016 over Trivandrum and Hyderabad.





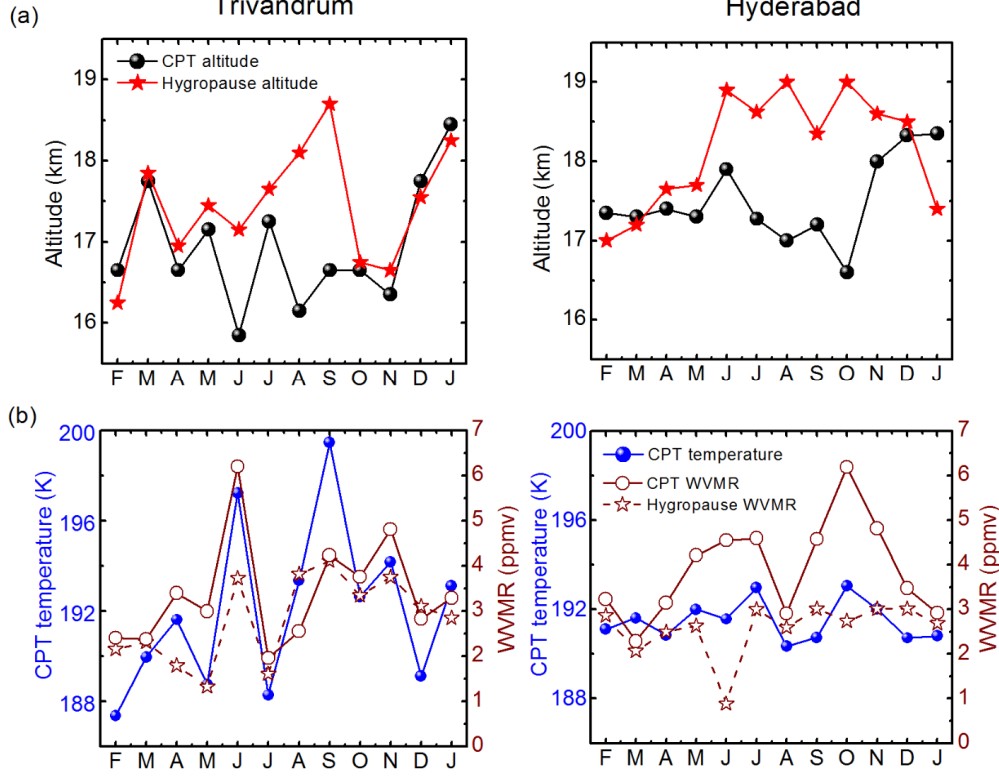

**Figure 5:**(a) Annual cycle of CPT altitude and hygropause altitude at Trivandrum and Hyderabad  (b) Annual cycle of CPT temperature, CPT mixing ratio and hygropause mixing ratio at  Trivandrum and Hyderabad obtained from radiosonde and CFH observations during the period February 2015- January 2016.




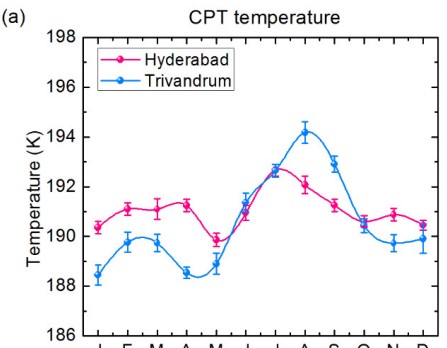

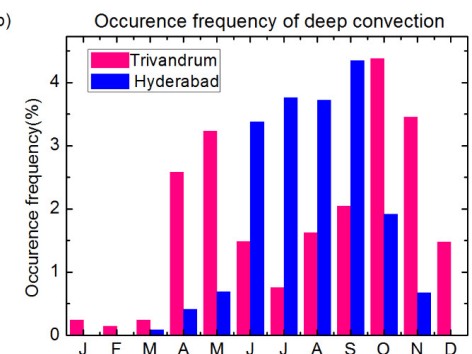

**Figure 6::** (a) Mean annual cycle of CPT temperature over Trivandrum and Hyderabad derived from COSMIC-RO data during the period 2011-2015 (b) Mean annual cycle of occurrence frequency of deep convection (Cloud top >12km) over Trivandrum and Hyderabad derived from KALPANA 1 -VHRR during the period 2011-2015.



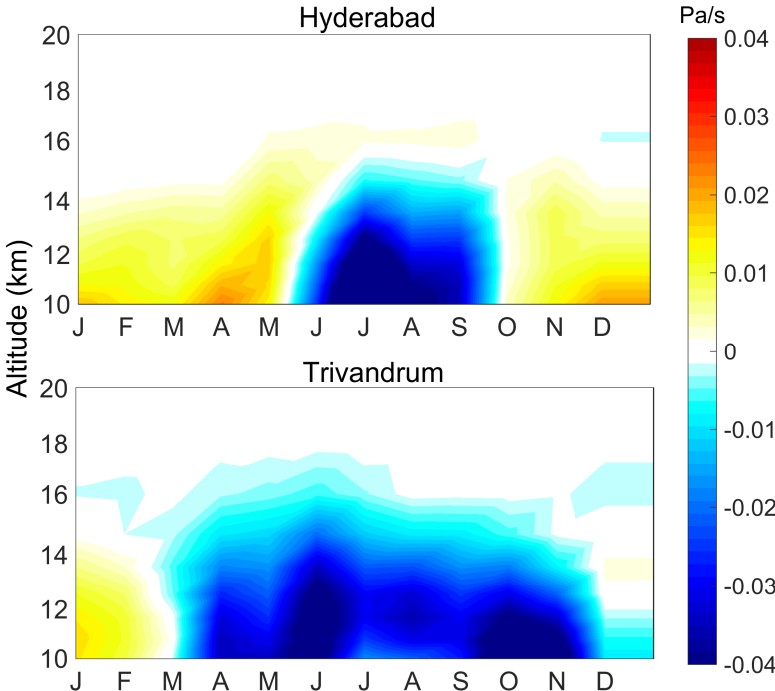

**Figure 7:** Mean annual cycle in ERA-interim vertical wind over Hyderabad and Trivandrum during the period 2011-2015.



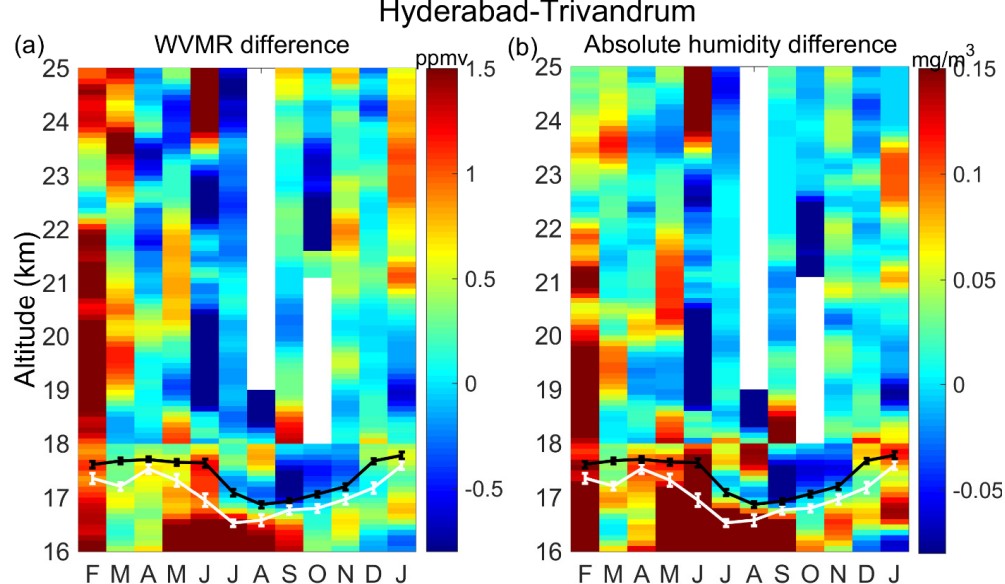

**Figure 8:** Annual cycle of (a) difference in WVMR and (b) difference in absolute humidity between Trivandrum and Hyderabad during the period February 2015- January 2016 derived from CFH observations. The difference in water vapour is taken by subtracting Trivandrum value from Hyderabad value. The line plots represents the mean CPT altitudes obtained from COSMIC-RO during the period 2011-2015 over Hyderabad (black) and Trivandrum (white).





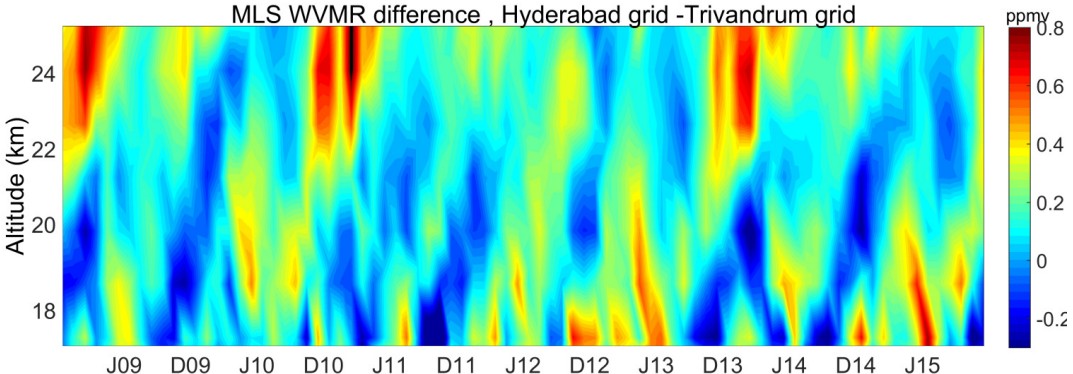

**Figure 9:** Annual cycle in difference of MLS obtained WVMR between Trivandrum grid and Hyderabad grid during the period 2009-2015. The difference in WVMR is taken by subtracting Trivandrum grid [6-11 ˚N; 72- 82 ˚E] from Hyderabad grid [15-20 ˚N; 73.5-83.5 ˚E].