# Peer review of "Annual cycle of water vapour in the lower stratosphere over the Indian Peninsula derived from Cryogenic Frost-point Hygrometer observations"

_Atmospheric Chemistry and Physics, 2018_

## Referee Comment (RC1) · Anonymous Referee #1 · 14 Sep 2018

General comments:

This study presents the water vapor profiles measured by the balloon-borne Cryogenic Frostpoint Hygrometer (CFH) in the upper troposphere and lower stratosphere (UTLS) over two stations in India during the period from February 2015 to January 2016. Their figures show that the CFH measurements have sufficient quality to discuss the nature of water vapor in the UTLS, in particular, the tape recorder signals observed at the two launching stations are very impressive. However, I think that the current manuscript lacks some essential and key points to understand and interpret the observational results. In my opinion, the required components are 1) employment of the saturation water vapor mixing ratio, 2) understanding a concept of three-dimensional transport in the UTLS, 3) presentation of the value to use the column integrated water vapor amount, and 4) presentation of the value to focus on the upward propagating signal in the water vapor mixing ratio difference between the two launching stations. The specific comments, including above four points, are described bellow.

Specific major comments:

1) The atmospheric pressure logarithmically changes with altitude. This is one of the reasons why we usually use the "mixing ratio" for our analysis because of its conservative property in vertical movement of the atmosphere. If one air parcel moves to upward, its air pressure, water vapor pressure, absolute humidity [mg/m^3] which the authors employ in the manuscript, must change, however, the water vapor mixing ratio never change without the occurrence of dehydration or hydration or mixing it with surrounding air mass. Therefore, when we want to discuss the water vapor and the dehydration, in particular in the tropical UTLS, we usually employ the minimum saturation water vapor mixing near the cold point tropopause (CPT), but not temperature at the CPT, to compare the observed water vapor mixing ratio. For example, here we consider two air parcels (parcel_1 and parcel_2), one has the temperature ($T\_1$) and pressure ($p\_1$) at altitude ($z\_1$), and another has ($T\_2$) and ($p\_2$) at ($z\_2$), and we assume parcel_1 locates higher altitude than parcel_2 ($z\_1 > z\_2$). If $T\_1$ and $T\_2$ are the same value, the two produce the same saturation water vapor mixing ratios ($p\_wv1$ and $p\_wv2$). However, the two situations produce different saturation water vapor mixing ratios ($SMR\_1$ and $SMR\_2$) because they are obtained from $SMR\_1 = p\_wv1/p\_1$ and $SMR\_2 = p\_wv2/p\_2$ under the condition of $p\_1 < p\_2$. This fact imposes the employment of the minimum SMR (SMRmin) near the CPT (the altitude where produces the SMRmin does not always agree with the CPT) on the current manuscript to discuss dehydration or hydration, in particular, in the following parts. Figure 1, Figure 3 (Could you include symbols showing the mean SMRmin at the altitude where they

produce in the same color scale to water vapor?), Figure 5b, Figure 6a, Discussions in Page5 Line27-Page6 Line2, Page7 Line20-30, the first paragraph in Page8, and Page11 Line21-24.

2) Though the authors cite some articles (e.g., Randel and Park, 2006; Park et al., 2007) addressing the Asian summer monsoon (ASM), a modern concept of the ASM is not sufficiently reflected in the interpretation of the results obtained from the current study. To grasp the concept, I think Figure 14 of Park et al. (2009) and Ploeger et al. (2017) may be helpful. They present the pictures involved in the ASM that consists rapid vertical transport by convections, horizontal transport by anticyclonic circulation at the UTLS, and slow ascent in the tropical stratosphere by the BDC. After considering those transport mechanisms involved in the ASM, I basically agree the interpretation that the water-rich air mass at higher altitude than that of the CPT observed over Hyderabad during ASM season, which might be transported from the region over Bay of Bengal (BoB) after it is hydrated by convections. It likely occurs, I think. But, if so, I think the infrared data around BoB (as well as other upstream regions of the anticyclonic circulation) should be additionally shown together with the horizontal wind field at just above the CPT altitude.

3) I could not find the reasonable reason why the authors employ the column integrated water vapor in the LS (IWV_LS) in the current manuscript. The IWV_LS is mainly discussed in the text in Page7 Line7-20 and the discussion about its difference between the two launching site is connected to local processes. I think it could not provide scientific discussions unless the concept of three-dimensional transport associated with the ASM is accurately introduced as described in the previous comment. On the other hand, in my opinion, if the authors successfully determine some indicator to quantify the hydration amount above the CPT altitude (strictly the SMRmin altitude) caused by local convection and/or ASM (for example, to calculate the vertical integration of the water "increment" from the local SMRmin, etc.) and if the observed water vapor profiles can be quantitatively interpreted in connection with hydration processes using the

indicator (for example, to show the relationship between the amount of the indicator and the ice water content in the convective overshooting clouds, etc.), such study may provide an new insight to understand the role of ASM on the stratospheric water vapor.

4) The authors focus on the upward propagating signal in the water vapor mixing ratio difference between the two launching stations in Figure 8. But I could not identify such propagating signal in the figure. On the other hand, Figure 9, indeed, clearly shows such upward propagating signal. This signal, however, can be simply produced by lager and smaller amplitudes of the tape recorder over Trivandrum and Hyderabad, respectively. Such interpretation is likely reasonable to me because Trivandrum locates nearer the center of the tropical pipe in the stratosphere than Hyderabad. How do you think about this opinion? You can check it by making some figures which show the meridional (latitude-altitude cross-section) distribution of water vapor mixing ratio over a meridian line across India (for example 80degE) for every month by using MLS data (like as Figure 1 in Ploeger et al., 2017).

References Park, M., W. J. Randel, L. K. Emmons, and N. J. Livesey (2009), Transport pathways of carbon monoxide in the Asian summer monsoon diagnosed from Model of Ozone and Related Tracers (MOZART), J. Geophys. Res., 114, D08303, doi:10.1029/2008JD010621.

Ploeger et al., Quantifying pollution transport from the Asian monsoon anticyclone into the lower stratosphere, Atmos. Chem. Phys., 17, 7055–7066, https://doi.org/10.5194/acp-17-7055-2017, 2017.

---

## Referee Comment (RC2) · Anonymous Referee #2 · 28 Sep 2018

This paper investigates the processes which control the water vapour budget at the seasonal scale in the tropical UTLS region above two Indian sites through the use of balloon-borne profiles of water vapour. The study is completed by space-borne observations of temperature and water vapour.

I do not think the manuscript adds much to the general knowledge of the processes explaining the seasonal and interannual control of the UTLS water vapour variations (connection with the tropopause temperature, tape recorder). It is not also a case study of dehydration or hydration effects. Although to me more investigations using statisti-

cal analysis from backtrajectory calculations (from the position of the balloon profiles) matching some locations of convective systems (through OLR) would have completed the study, the paper is nice and interestingly addresses the effects of dynamical processes (convection, horizontal transport in the UTLS and BDC in the stratosphere) in the specific Indian subcontinent impacted by strong seasonal dynamical, microphysical and chemical variations associated with the monsoon in summer. Also, one of the most positive point of the presented work is that it definitely promotes the precious need for regular in situ water vapour observations in the tropical UTLS and stratosphere for all seasons in Asia, with India as a highly valuable location for such investigations.

The balloon records presented here would be helpful to decrease the uncertainty in model convective parameterization and in currently assimilated in the reanalysis modeling systems which typically show some errors in UTLS water vapour due to the limitations of observational data in the tropics, especially for the Asian monsoon season.

The idea to use water vapour observations at stratospheric altitudes from both Indian sites as a proxy for the ascent rate of BDC is interesting given the possibility of regular balloon launches from these locations that is to be supported by the scientific community. Also the paper is very nice to read and well concise. I would recommend publication in ACP after the following comments have been addressed.

General comments

One point would be to investigate if the features observed on the balloon-borne profiles over both stations are typical or repetitive in this region on a seasonal (and perhaps interannual) scale. The authors interpret the vertical dependence of WVMR (especially in fig.2 and chapter 3.1) through general concepts of transport. Have they attempted to verify this statements using some satellite data (MLS/Aura mainly)? For instance, the explanation dealing with synoptic variabilities for the minimum of WVMR in summer around 21 km could be checked from seasonal variations inferred from MLS over the same period (MLS are anyway used in this study to investigate differences between

both sites in Fig.9). In other words, one would have appreciated a better inclusion of satellite observations in the interpretation of the vertical profiles on Fig. 1 and 2. I think the manuscript should better address the possibility of local effects to explain differences observed from the CFH balloon records over the Hyderabad and Trivandrum locations. The authors describe the seasonal difference on the WVMR profiles between Hyderabad and Trivandrum (through Fig.8). The problem is that I do not see the features described in chapter 3.3 (especially, I cannot verify the statement that the WVMR just above the CPT altitude, around 17-18 km, is relatively high over Hyderabad during summer monsoon (JJAS) and winter seasons (DJF) and high over Trivandrum during pre-monsoon (MAM) and post-monsoon (ON) seasons, also from Fig.3). The propagation of the water vapour amount difference similar to a tape recorder shape is also not obvious at all to me. However, features are more apparent when MLS WVMR is used to highlight the differences (Fig.9). To me, the different features observed from CFH and MLS in terms of WVMR may reflect significant local effects controlling the water vapour budget on both sites whereas the use of a 5°x10° grid tends to smooth out the effects. I do not think the differences to be caused by the measurement quality because as pointed out by the authors their amplitude is higher than instrumental uncertainties. Could local effects be due to local convection impacting the water vapour budget? However, something striking is that the "noisy" differences are visible up to 25 km and not only near CPT whatever the season (not only in convective seasons). Could this be due to long-range transport of hydrated or dehydrated air masses? What do the authors think about this? I would recommend the authors to clarify, simplify or remove this part (P9 lines 16-30 and/or Fig.8). Same remark for the Summary/conclusion part.

Specific comments:

P2 Line 3: "Due to the large residence time (of more than a year) stratospheric water vapour contributes significantly to the climate forcing instead of a simple response (Wang et al., 2009)" What do the authors mean by "simple response"? Please clarify the end of the sentence. P2 line 8: you mean direct injection of water vapour by

volcanic eruptions? P3 line 6: you can add 2 other references to span other balloon-borne FP hygrometers than the typical NOAA CFH: Vömel, H., V. Yushkov, S. Khaykin, L. Korshunov, E. Kyrö, and R. Kivi, Intercomparisons of tratospheric Water Vapor Sensors: FLASH-B and NOAA/CMDL Frost-Point Hygrometer, Journal of Atmospheric and Oceanic Technology, Vol.24, 941-952, 2007.

Mélanie Ghysels, Emmanuel D. Riviere, Sergey Khaykin, Clara Stoeffler, Nadir Amarouche, Jean-Pierre Pommereau, Gerhard Held, and Georges Durry, Inter-comparison of in situ water vapor balloon-borne measurements from Pico-SDLA H2O and FLASH-B in the tropical UTLS, Atmos. Meas. Tech., 9, 1207-1219, https://doi.org/10.5194/amt-9-1207-2016, 2016.

Berthet, G., J.-B. Renard, M. Ghysels, G. Durry, B. Gaubicher and N. Amarouche, Balloon-borne observations of mid-latitude stratospheric water vapour: comparisons with HALOE and MLS satellite data, J. Atmos. Chem., 70:197-219, doi: 10.1007/10874-013-9264-7, 2013.

P4 line 7: in the sentence "MLS provides water vapour profiles with a vertical resolution of 2-3 km, 4-6 km and 8 km at 316 hPa to tropopause, tropopause to 1 hPa and at 0.1 hPa with precisions of ∼15 %, ∼0.1 ppmv and ∼0.3 ppmv respectively" it is not clear to which altitude range the "respectively" term corresponds. 316 hPa to tropopause? tropopause to 1 hPa? at 0.1 hPa? Please clarify.

P3 line 24: It would be appreciable if the authors could discuss the choice of ERA-I re-analysis system keeping in mind the reported differences between reanalysis (ERA-I, MERRA, MERRA2, JRA) in tropical UTLS dynamics or at least provide relevant references quantifying these differences.

Figure 1: the minimum and maximum temperature values on the abscissa axis should be 180 K and 240 K respectively so that the reader can better distinguish differences between profiles.

P5 line 22: I do not see why the authors discuss the peak of ∼20 ppmv at 15 km in the manuscript which is focused on processes for levels above. Either the origin of the peak is discussed here (link to convection outflow for instance) or please remove it.

P7 lines 22-30: the authors do not comment the reason for the minimum in CPT in summer (July) both in Hyderabad and Trivandrum which values are comparable to the ones in winter. Is it a local effect?

P8 lines 1-15: Similar question here: I agree that in general the mean annual pattern in CPT temperature is consistent with the annual pattern of WVMR obtained from CFH observations at CPT (as seen from Fig. 5) but why the CPT seasonal variation derived from COSMIC (Fig. 6) differs from that observed from the balloons (Fig. 5) with no minimum in July in the COSMIC time series? Is it due to local effects or different periods used? Please explain this in the text.

P9 lines 6-7: the authors state: "Though the magnitude of vertical wind in any reanalysis may not be very accurate, the direction (updraft/ downdraft) would be quite reliable". Could you cite some references or studies having addressed this issue?

P9 line 27: I think the sentence "difference in WVMR could be due to the difference in pressure also since it is the ratio of vapour pressure to the atmospheric pressure" could be checked easily from pressure profiles or fields above both sites. Why do the authors did not investigate the (not very probable with respect to hydration or dehydration effects impact water vapour absolute concentration) pressure variations?

Legend of Fig.9: please specify to which months "J" and "D" correspond.

Technical corrections

P2 line 8: Please write "the tropopause" P2 line 12: You should define the BDC acronym in the abstract P2 line 14: the LS acronym is already defined in the abstract P2 line27: please define the SST acronym (Sea Surface Temperature)

---

## Author Comment (AC1) · 28 Nov 2018

General comments:

This study presents the water vapor profiles measured by the balloon-borne Cryogenic Frost point Hygrometer (CFH) in the upper troposphere and lower stratosphere (UTLS) over two stations in India during the period from February 2015 to January 2016. Their figures show that the CFH measurements have sufficient quality to discuss the nature of water vapor in the UTLS, in particular, the tape recorder signals observed at the two launching stations are very impressive. However, I think that the current manuscript lacks some essential and key points to understand and interpret the observational results. In my opinion, the required components are 1) employment of the saturation water vapor mixing ratio, 2) understanding a concept of three-dimensional transport in the UTLS, 3) presentation of the value to use the column integrated water vapor amount, and 4) presentation of the value to focus on the upward propagating signal in the water vapor mixing ratio difference between the two launching stations. The specific comments, including above four points, are described below.

**Response:** First of all, we thank the anonymous referee for the appreciation and valuable comments. We have taken into account all the comments and suggestions in preparing the revised manuscript. The response to each specific comment is given below.

Specific major comments:

1)       The atmospheric pressure logarithmically changes with altitude. This is one of the reasons why we usually use the "mixing ratio" for our analysis because of its conservative property in vertical movement of the atmosphere. If one air parcel moves to upward, its air pressure, water vapor pressure, absolute humidity [mg/mˆ3] which the authors employ in the manuscript, must change, however, the water vapor mixing ratio never change without the occurrence of dehydration or hydration or mixing it with surrounding air mass. Therefore, when we want to discuss the water vapor and the dehydration, in particular in the tropical UTLS, we usually employ the minimum saturation water vapor mixing near the cold point tropopause (CPT), but not temperature at the CPT, to compare the observed water vapor mixing ratio. For example, here we consider two air parcels (parcel_1 and parcel_2), one has the temperature ($T_1$) and pressure ($p_1$) at altitude ($z_1$), and another has ($T_2$) and ($p_2$) at ($z_2$), and we assume parcel_1 locates higher altitude than parcel_2 ($z_1 > z_2$). If $T_1$ and $T_2$ are the same value, the two produce the same saturation water vapor mixing ratios ($p_{wv1}$ and $p_{wv2}$). However, the two situations produce different saturation water vapor mixing ratios ($SMR_1$ and $SMR_2$) because they are obtained from $SMR_1 = p_{wv1}/p_1$ and $SMR_2 = p_{wv2}/p_2$ under the condition of $p_1 < p_2$. This fact imposes the employment of the minimum SMR (SMRmin) near the CPT (the altitude where produces the SMRmin does not always agree with the CPT) on the current manuscript to discuss dehydration or hydration, in particular, in the following parts. Figure 1, Figure 3 (Could you include symbols showing the mean SMRmin at the altitude where they produce in the same color scale to water vapor?), Figure 5b, Figure 6a, Discussions in Page5 Line27-Page6 Line2, Page7 Line20-30, the first paragraph in Page8, and Page11 Line21-24.

**Response:** The suggestion is well taken. The SMRmin values are estimated and is included in Figure 1. The mean annual variation of SMRmin altitude is shown in Figure 3 along with the monthly variation of CPT altitude. The annual variation of SMRmin altitude and SMRmin from CFH observations are added in Figure 5. The mean annual cycle of SMRmin value is added

in Figure 6a along with the CPT temperature. The SMRmin altitude occurs within 500 m below the CPT altitude and the difference between these two altitudes is maximum in the winter months. The discussions are made in the revised manuscript accordingly.

2)      Though the authors cite some articles (e.g., Randel and Park, 2006; Park et al., 2007) addressing the Asian summer monsoon (ASM), a modern concept of the ASM is not sufficiently reflected in the interpretation of the results obtained from the current study. To grasp the concept, I think Figure 14 of Park et al. (2009) and Ploeger et al. (2017) may be helpful. They present the pictures involved in the ASM that consists rapid vertical transport by convections, horizontal transport by anticyclonic circulation at the UTLS, and slow ascent in the tropical stratosphere by the BDC. After considering those transport mechanisms involved in the ASM, I basically agree the interpretation that the water-rich air mass at higher altitude than that of the CPT observed over Hyderabad during ASM season, which might be transported from the region over Bay of Bengal (BoB) after it is hydrated by convections. It likely occurs, I think. But, if so, I think the infrared data around BoB (as well as other upstream regions of the anticyclonic circulation) should be additionally shown together with the horizontal wind field at just above the CPT altitude.

**Response:** As suggested, spatial distribution of occurrence of deep convection (using thermal infrared data), horizontal wind field (using ERA-Interim reanalysis data) and potential path ways of air mass (using HYSPLIT transport model back trajectories) are generated for different seasons and is shown as new figures (Figures 6 and 7) in the revised version). Vertical wind already in the manuscript. As suggested, three-dimensional transport and hydration of LS during ASM period are discussed.

3)      I could not find the reasonable reason why the authors employ the column integrated water vapor in the LS (IWV_LS) in the current manuscript. The IWV_LS is mainly discussed in the text in Page7 Line7-20 and the discussion about its difference between the two launching site is connected to local processes. I think it could not provide scientific discussions unless the concept of three-dimensional transport associated with the ASM is accurately introduced as described in the previous comment. On the other hand, in my opinion, if the authors successfully determine some indicator to quantify the hydration amount above the CPT altitude (strictly the SMRmin altitude) caused by local convection and/or ASM (for example, to calculate the vertical integration of the water "increment" from the local SMRmin, etc.) and if the observed water vapor profiles can be quantitatively interpreted in connection with hydration processes using the indicator (for example, to show the relationship between the amount of the indicator and the ice water content in the convective overshooting clouds, etc.), such study may provide an new insight to understand the role of ASM on the stratospheric water vapor.

**Response:** The concept of three-dimensional transport is discussed in the revised manuscript. The spatial variation of deep convection, horizontal and vertical transport and their effect on the Lower Stratosphere (LS) water vapour are discussed in the revised manuscript. The seasonal variability of LS water vapour (Figure 2) shows large variability in the altitude region CPT-21 km compared to the altitude region 21-25 km. Hence, Figure 4 (annual variation of $IWV_{LS}$) is modified in the revised manuscript. The CPT to 25 km region (LS) was separated into two regimes, viz CPT-21 km (LS1) & 21-25 km (LS2). IWV in lower regime, LS1 is influenced directly by local/regional tropospheric dynamics and contributes about 50-70 % of the $IWV_{LS}$. Hence, the variability of IWV (from the annual mean) in this regime can be used as an indicator for quantifying/understanding the amount of water vapour entered in to the lower stratosphere from convective disturbances/monsoon dynamics. The integrated water vapour in the altitude region 21-25 km, is approximately 30-40 % of the total $IWV_{LS}$ shows similar variability over both the stations are controlled mainly by large scale dynamics.

4)      The authors focus on the upward propagating signal in the water vapor mixing ratio difference between the two launching stations in Figure 8. But I could not identify such propagating signal in the

figure. On the other hand, Figure 9, indeed, clearly shows such upward propagating signal. This signal, however, can be simply produced by larger and smaller amplitudes of the tape recorder over Trivandrum and Hyderabad, respectively. Such interpretation is likely reasonable to me because Trivandrum locates nearer the centre of the tropical pipe in the stratosphere than Hyderabad. How do you think about this opinion? You can check it by making some figures which show the meridional (latitude-altitude cross-section) distribution of water vapor mixing ratio over a meridian line across India (for example 80degE) for every month by using MLS data (like as Figure 1 in Ploeger et al., 2017).

**Response:** As pointed by the reviewer, the difference between the two stations and its propagation is not clearly visible from the CFH observations; This could be mainly due to the local effects such the day-to-day variability in CPT temperature and convection and/or due to the usage of lesser number of profiles in each month (1or 2 profiles). Also note that CFH provides better vertical resolution. The signature of upward propagation in Figure 8 could be improved if smoothed for atleast 1 km. But, it is equal to degrading the vertical resolution of CFH observations. In the revised version we have applied a 3-point smoothing for better representation and the upward propagation is marked with an arrow mark.
As rightly pointed out, the higher ascent rate at Trivandrum is expected as it is an equatorial station. Hyderabad being an off-equatorial station, the vertical ascent is relatively small compared Trivandrum. As suggested, meridional (latitude-altitude cross-section) distribution of water vapor mixing ratio over a meridian line across India (75- 80 °E) is generated for every month using MLS data, for examining the latitudinal differences in amplitude of signals and is included as a Figure in the revised manuscript (Figure 12 in revised version). The latitudinal differences in the water vapour signal is discussed in the revised text in section 3.3.

References Park, M., W. J. Randel, L. K. Emmons, and N. J. Livesey (2009), Transport pathways of carbon monoxide in the Asian summer monsoon diagnosed from Model of Ozone and Related Tracers (MOZART), J. Geophys. Res., 114, D08303, doi: 10.1029/2008JD010621.

Ploeger et al., (2017), Quantifying pollution transport from the Asian monsoon anticyclone into the lower stratosphere, Atmos. Chem. Phys., 17, 7055–7066, https://doi.org/10.5194/acp-17-7055-2017.

**Response:** The references are noted and cited in the revised manuscript.

**Once again, we thank the reviewer for the constructive comments**
* * *

---

## Author Comment (AC2) · 28 Nov 2018

This paper investigates the processes which control the water vapour budget at the seasonal scale in the tropical UTLS region above two Indian sites through the use of balloon-borne profiles of water vapour. The study is completed by space-borne observations of temperature and water vapour.

I do not think the manuscript adds much to the general knowledge of the processes explaining the seasonal and interannual control of the UTLS water vapour variations (connection with the tropopause temperature, tape recorder). It is not also a case study of dehydration or hydration effects. Although to me more investigations using statistical analysis from back trajectory calculations (from the position of the balloon profiles) matching some locations of convective systems (through OLR) would have completed the study, the paper is nice and interestingly addresses the effects of dynamical processes (convection, horizontal transport in the UTLS and BDC in the stratosphere) in the specific Indian subcontinent impacted by strong seasonal dynamical, microphysical and chemical variations associated with the monsoon in summer. Also, one of the most positive point of the presented work is that it definitely promotes the precious need for regular in situ water vapour observations in the tropical UTLS and stratosphere for all seasons in Asia, with India as a highly valuable location for such investigations. The balloon records presented here would be helpful to decrease the uncertainty in model convective parameterization and in currently assimilated in the reanalysis modelling systems which typically show some errors in UTLS water vapour due to the limitations of observational data in the tropics, especially for the Asian monsoon season.

The idea to use water vapour observations at stratospheric altitudes from both Indian sites as a proxy for the ascent rate of BDC is interesting given the possibility of regular balloon launches from these locations that is to be supported by the scientific community.

Also the paper is very nice to read and well concise. I would recommend publication in ACP after the following comments have been addressed.

Response: First of all, we thank the anonymous referee for his appreciation and valuable comments. We have taken into account all the comments and suggestions in preparing the revised manuscript. Three-dimensional concept of water vapour transport in the UTLS regions has been discussed using the occurrence frequency of deep convection from thermal Infrared data, horizontal and vertical transport from wind field and potential path ways of air mass (back trajectory analysis) for different seasons and is shown as new figures (Figures 6 and 7 in the revised version).

The response to each specific comment is given below.

General comments

One point would be to investigate if the features observed on the balloon-borne profiles over both stations are typical or repetitive in this region on a seasonal (and perhaps interannual) scale. The authors interpret the vertical dependence of WVMR (especially in fig.2 and chapter 3.1) through general concepts of transport. Have they attempted to verify this statements using some satellite data (MLS/Aura mainly)? For instance, the explanation dealing with synoptic variabilities for the minimum of WVMR in summer around 21 km could be checked from seasonal variations inferred from MLS over the same period (MLS are anyway used in this study to investigate differences between both sites in

Fig.9). In other words, one would have appreciated a better inclusion of satellite observations in the interpretation of the vertical profiles on Fig. 1 and 2.

**Response:** The features observed on the balloon-borne profiles in Figure 1 over both the stations are typical in the region. The seasonal mean water vapour profiles using MLS data also shows almost similar seasonal patterns to that obtained using CFH observations. However, MLS profiles appear smoother than the CFH profiles partly due to lesser vertical resolution and partly due to greater number of profiles. The MLS derived seasonal mean profiles for the year 2015 is added in Figure 2 of the revised version.

I think the manuscript should better address the possibility of local effects to explain differences observed from the CFH balloon records over the Hyderabad and Trivandrum locations. The authors describe the seasonal difference on the WVMR profiles between Hyderabad and Trivandrum (through Fig.8). The problem is that I do not see the features described in chapter 3.3 (especially, I cannot verify the statement that the WVMR just above the CPT altitude, around 17-18 km, is relatively high over Hyderabad during summer monsoon (JJAS) and winter seasons (DJF) and high over Trivandrum during pre-monsoon (MAM) and post-monsoon (ON) seasons, also from Fig.3). The propagation of the water vapour amount difference similar to a tape recorder shape is also not obvious at all to me. However, features are more apparent when MLS WVMR is used to highlight the differences (Fig.9). To me, the different features observed from CFH and MLS in terms of WVMR may reflect significant local effects controlling the water vapour budget on both sites whereas the use of a 5_x10_ grid tends to smooth out the effects. I do not think the differences to be caused by the measurement quality because as pointed out by the authors their amplitude is higher than instrumental uncertainties. Could local effects be due to local convection impacting the water vapour budget? However, something striking is that the "noisy" differences are visible up to 25 km and not only near CPT whatever the season (not only in convective seasons). Could this be due to long-range transport of hydrated or dehydrated air masses? What do the authors think about this? I would recommend the authors to clarify, simplify or remove this part (P9 lines 16-30 and/or Fig.8). Same remark for the Summary/conclusion part.

**Response:** As pointed out, the difference in water vapour amount between the two stations and its propagation is not clearly visible in CFH observations. This could be mainly due to the local effects such the day-to-day variability in CPT temperature and convection and/or due to the usage of lesser number of profiles in each month (1or 2 profiles). In the MLS WVMR (Figure 9) the difference is clearly seen. Figure 8 could be improved if profiles are smoothed for atleast 1 km. But, it is equal to degrading the vertical resolution of CFH observations. In the revised version, we have applied a 3-point smoothing to the water vapour difference profiles for better representation. The upward propagation is clearer now and marked with an arrow mark. The discussion in the manuscript is also modified.

In order to address the local effects on water vapour distribution in LS, Figure 4(annual variation of $IWV_{LS}$) is modified in the revised manuscript. The CPT to 25 km region (LS) was separated into two regimes, viz CPT-21 km (LS1) & 21-25 km (LS2). IWV in lower regime, CPT- 21 km region which contributes about 60-70 % of the $IWV_{LS}$ indicates the direct effect of local/regional dynamics. But, the integrated water vapour in the LS2 region does not show much variability and have no direct association with regional/local dynamics. Hence, we feel that the noisy differences upto 25 km could not be attributed completely to the deep convection. The direct influence of deep convection can be seen up to a maximum altitude of 20-21 km; that is the LS1 region. Above that region, the large-scale dynamics (BDC) mainly controls the transport of water vapour. The influence of methane oxidation and long-range advection also may play a role in this altitude region.

Specific comments:

P2 Line 3: "Due to the large residence time (of more than a year) stratospheric water vapour contributes significantly to the climate forcing instead of a simple response (Wang et al., 2009)" What do the authors mean by "simple response"? Please clarify the end of the sentence.

**Response:** Simple response refer to instantaneous direct effect But, as it seems to be confusing we have deleted 'instead of a simple response' in the revised manuscript.

P2 line 8: you mean direct injection of water vapour by volcanic eruptions?

**Response:** Yes.

P3 line 6: you can add 2 other references to span other balloon borne FP hygrometers than the typical NOAA CFH:

Vömel, H., V. Yushkov, S. Khaykin, L. Korshunov, E. Kyrö, and R. Kivi, Intercomparisons of stratospheric Water Vapor Sensors: FLASH-B and NOAA/CMDL Frost-Point Hygrometer, Journal of Atmospheric and Oceanic Technology, Vol.24, 941-952, 2007.

Mélanie Ghysels, Emmanuel D. Riviere, Sergey Khaykin, Clara Stoeffler, Nadir Amarouche, Jean-Pierre Pommereau, Gerhard Held, and Georges Durry, Intercomparison of in situ water vapor balloon-borne measurements from Pico-SDLA H2O and FLASH-B in the tropical UTLS, Atmos. Meas. Tech., 9, 1207-1219, https://doi.org/10.5194/amt-9-1207-2016, 2016.
Berthet, G., J.-B. Renard, M. Ghysels, G. Durry, B. Gaubicher and N. Amarouche, Balloon-borne observations of mid-latitude stratospheric water vapour: comparisons with HALOE and MLS satellite data, J. Atmos. Chem., 70:197-219, doi: 10.1007/10874-013-9264-7, 2013.

**Response:** Suggested references added in the revised version.

P4 line 7: in the sentence "MLS provides water vapour profiles with a vertical resolution of 2-3 km, 4-6 km and 8 km at 316 hPa to tropopause, tropopause to 1 hPa and at 0.1 hPa with precisions of _15 %, _0.1 ppmv and _0.3 ppmv respectively" it is not clear to which altitude range the "respectively" term corresponds. 316 hPa to tropopause? Tropopause to 1 hPa? At 0.1 hPa? Please clarify.

**Response:** The sentence is split into two sentences for more clarity.

P3 line 24: It would be appreciable if the authors could discuss the choice of ERA-I reanalysis system keeping in mind the reported differences between reanalysis (ERA-I, MERRA, MERRA2, JRA) in tropical UTLS dynamics or at least provide relevant references quantifying these differences.

**Response:** Though there are differences between the different reanalysis, all the datasets shows almost similar feature/pattern. There are several Intercomparison between different reanalysis data. SPARC Intercomparison of Middle Atmosphere Climatologies (SPARC, 2002; Randel et al., 2004) have inter compared reanalyses and related data sets in the middle atmosphere. Recently, the Stratosphere–troposphere Processes And their Role in Climate (SPARC) Reanalysis Intercomparison Project (S-RIP) compared the reanalysis data sets using a variety of key diagnostics (Fujiwara et al. 2017). In the revised manuscript, the choice of ERA-I reanalysis is discussed by citing appropriate references.

Figure 1: the minimum and maximum temperature values on the abscissa axis should be 180 K and 240 K respectively so that the reader can better distinguish differences between profiles.

**Response:** Suggestion incorporated

P5 line 22: I do not see why the authors discuss the peak of _20 ppmv at 15 km in the manuscript which is focused on processes for levels above. Either the origin of the peak is discussed here (link to convection outflow for instance) or please remove it.

**Response:** Text modified.

P7 lines 22-30: the authors do not comment the reason for the minimum in CPT in summer (July) both in Hyderabad and Trivandrum which values are comparable to the ones in winter. Is it a local effect?

**Response:** The minimum in CPT in summer (July) was due to the effect local/regional deep convection. This is discussed in the revised text.

P8 lines 1-15: Similar question here: I agree that in general the mean annual pattern in CPT temperature is consistent with the annual pattern of WVMR obtained from CFH observations at CPT (as seen from Fig. 5) but why the CPT seasonal variation derived from COSMIC (Fig. 6) differs from that observed from the balloons (Fig. 5) with no minimum in July in the COSMIC time series? Is it due to local effects or different periods used? Please explain this in the text.

**Response:** The difference in CPT in July (from balloon flights) is due to the effect of local deep convection. The dehydration effect is seen in the MLS also. This is not evident in the COSMIC profile as it is the mean for five years (2011-2015). The convective effect on CPT and water vapour has inter-annual variability also. Clarity is made in the revised text.

P9 lines 6-7: the authors state: "Though the magnitude of vertical wind in any reanalysis may not be very accurate, the direction (updraft/ downdraft) would be quite reliable". Could you cite some references or studies having addressed this issue?

**Response:** The vertical wind in ERA-interim reanalysis is calculated from the convergence/ divergence of the archived u and v fields on interpolated pressure levels (eg: Sanz et al., 2007; Wohltmann and Rex, 2008). And, hence the direction of the wind will be quite reliable even though the magnitude may be erroneous. The ERA-interim reanalysis uses a w-equation balance operator in the background constraint (Fisher, 2003). Ploeger et al 2010 from studies of transport in the TTL has found that though the transport characteristics are depending on the vertical velocity, robust patterns of transport in the TTL have greater reliability than in exact numbers. The text in revised version is modified and cited the above said references.

P9 line 27: I think the sentence "difference in WVMR could be due to the difference in pressure also since it is the ratio of vapour pressure to the atmospheric pressure" could be checked easily from pressure profiles or fields above both sites. Why do the authors did not investigate the (not very probable with respect to hydration or dehydration effects impact water vapour absolute concentration) pressure variations?

**Response:** The difference in WVMR between the stations (Figure 8) is due to difference in actual amount of water vapour itself. Otherwise, it wouldn't have seen in the difference in absolute humidity between the stations (Absolute humidity doesn't have pressure dependence).

Legend of Fig.9: please specify to which months "J" and "D" correspond.

**Response:** J corresponds to June and D corresponds to December. It is added in the revised Figure caption of the mentioned Figure.

Technical corrections P2 line 8: Please write "the tropopause"

**Response:** Corrected

 P2 line 12: You should define the BDC acronym in the abstract P2 line 14: the LS acronym is already defined in the abstract

**Response:** Corrected

P2 line27: please define the SST acronym (Sea Surface Temperature)

**Response:** Corrected

**Once again, we thank the reviewer for the constructive comments**
**\*\*\*\*\*\*\*\***